# Solid Lipid Nanoparticles Containing Dopamine and Grape Seed Extract: Freeze-Drying with Cryoprotection as a Formulation Strategy to Achieve Nasal Powders

**DOI:** 10.3390/molecules28237706

**Published:** 2023-11-22

**Authors:** Elvira De Giglio, Udo Bakowsky, Konrad Engelhardt, Antonello Caponio, Matteo La Pietra, Stefania Cometa, Stefano Castellani, Lorenzo Guerra, Giuseppe Fracchiolla, Maria Luana Poeta, Rosanna Mallamaci, Rosa Angela Cardone, Stefano Bellucci, Adriana Trapani

**Affiliations:** 1Department of Chemistry, University of Bari “Aldo Moro”, 70125 Bari, Italy; elvira.degiglio@uniba.it; 2Department of Pharmaceutics and Biopharmaceutics, Philipps University of Marburg, Robert-Koch-Str. 4, 35037 Marburg, Germany; ubakowsky@aol.com (U.B.); konrad.engelhardt@pharmazie.uni-marburg.de (K.E.); 3Department of Pharmacy-Drug Sciences, University of Bari “Aldo Moro”, Via Orabona 4, 70125 Bari, Italy; antonello.caponio@gmail.com (A.C.); giuseppe.fracchiolla@uniba.it (G.F.); 4Istituto Nazionale di Fisica Nucleare-Laboratori Nazionali di Frascati, Via Enrico Fermi 54, 00044 Frascati, Italy; matteo.lapietra.97@gmail.com (M.L.P.); stefano.bellucci@lnf.infn.it (S.B.); 5Department of Information Engineering, Polytechnic University of Marche, 60131 Ancona, Italy; 6Jaber Innovation s.r.l., 00144 Rome, Italy; stefania.cometa@jaber.it; 7Department of Precision and Regenerative Medicine and Ionian Area (DiMePRe-J), University of Bari “Aldo Moro”, 70125 Bari, Italy; stefano.castellani@uniba.it; 8Department of Biosciences, Biotechnologies and Environment, University of Bari “Aldo Moro”, 70125 Bari, Italy; lorenzo.guerra1@uniba.it (L.G.); marialuana.poeta@uniba.it (M.L.P.); rosanna.mallamaci@uniba.it (R.M.); rosaangela.cardone@uniba.it (R.A.C.)

**Keywords:** dopamine, grape seed extract, cryoprotectants, atomic force microscopy, Raman spectroscopy, RPMI 2650 cell model line

## Abstract

(1) Background: DA-Gelucire^®^ 50/13-based solid lipid nanoparticles (SLNs) administering the neurotransmitter dopamine (DA) and the antioxidant grape-seed-derived proanthocyanidins (grape seed extract, GSE) have been prepared by us in view of a possible application for Parkinson’s disease (PD) treatment. To develop powders constituted by such SLNs for nasal administration, herein, two different agents, namely sucrose and methyl-β-cyclodextrin (Me-β-CD), were evaluated as cryoprotectants. (2) Methods: SLNs were prepared following the melt homogenization method, and their physicochemical features were investigated by Raman spectroscopy, Scanning Electron Microscopy (SEM), atomic force microscopy (AFM) and X-ray Photoelectron Spectroscopy (XPS). (3) Results: SLN size and zeta potential values changed according to the type of cryoprotectant and the morphological features investigated by SEM showed that the SLN samples after lyophilization appear as folded sheets with rough surfaces. On the other hand, the AFM visualization of the SLNs showed that their morphology consists of round-shaped particles before and after freeze-drying. XPS showed that when sucrose or Me-β-CD were not detected on the surface (because they were not allocated on the surface or completely absent in the formulation), then a DA surfacing was observed. In vitro release studies in Simulated Nasal Fluid evidenced that DA release, but not the GSE one, occurred from all the cryoprotected formulations. Finally, sucrose increased the physical stability of SLNs better than Me-β-CD, whereas RPMI 2650 cell viability was unaffected by SLN-sucrose and slightly reduced by SLN-Me-β-CD. (4) Conclusions: Sucrose can be considered a promising excipient, eliciting cryoprotection of the investigated SLNs, leading to a powder nasal pharmaceutical dosage form suitable to be handled by PD patients.

## 1. Introduction

Undoubtedly, in the field of neurodegenerative diseases, such as Alzheimer’s disease and Parkinson’s disease (PD), several issues in overcoming the Blood–Brain Barrier (BBB) are to be faced. The protection role exerted by the BBB mainly consists of limiting the access of xenobiotics into the brain compartment and, consequently, low bioavailability and limited brain penetration of administered drugs occurs. It prompts us to find alternative strategies to reach the Central Nervous System (CNS) in a comfortable way for patients affected by neurological disorders.

Currently, for patients affected by neurological disorders, a very promising approach is constituted by intranasal drug administration since the olfactory mucosa is the region where the BBB is interrupted. The unique characteristics of the nasal mucosa enable it to bypass the BBB, while its CNS connections allow drug transport directly into the brain [1,2,3]. In fact, molecule absorption through the trigeminal and olfactory nerves from the nasal cavity provides a direct drug entrance to the brain as well as decreased accumulation of therapeutic agents in the organs, such as the liver, spleen and kidney and, hence, reduced systemic side effects [4]. Additionally, intranasal administration is essentially painless; it requires a shorter time to onset of effect and higher bioavailability due to avoidance of hepatic first-pass metabolism. On the other hand, pharmaceutical dosage forms for intranasal administration are also needle-free and suitable for self-medication. However, nose-to-brain delivery has some limitations to be taken into account when this approach is pursued for the treatment of neurological disorders. Such drawbacks are related to the limited volume of each nostril in humans and, moreover, to the mucociliary clearance, which diminishes the residence times of formulations within the nasal cavity and consequent decrease in drug absorption. Hence, by prolonging nasal residence time and/or the attainment of higher local drug concentration, nose-to-brain delivery improves [5]. Besides this, it has been clarified that the application of nano-structured delivery systems involves further advantages from a nose-to-brain delivery point of view, including drug protection, release kinetic and drug absorption [5].

Among different types of colloidal carriers for pharmaceutical applications, solid lipid nanoparticles (SLNs) have been extensively investigated for administration through the nasal compartment and also for the treatment of PD [6,7,8]. In general, SLNs are recognized for their low inherent toxicity and for their economically advantageous large-scale production. Moreover, SLNs are capable of providing sustained release and are also able to encapsulate both hydrophobic and hydrophilic active principles. In this regard, we have demonstrated that Gelucire^®^ 50/13, a self-emulsifying lipid, as a component of SLNs, is capable of increasing the drug-loading of hydrophilic active principles [9,10] such as the neurotransmitter dopamine (DA), whose concentration is diminished in the Parkinsonian patient’s brain. Therefore, we have prepared DA-loaded Gelucire^®^ 50/13 SLNs (DA-SLNs) [11] and, then, DA-SLN adsorbing grape seed extract (GSE) SLNs (i.e., GSE/DA-SLNs) by the melt homogenization method [11] and, then, DA-SLN adsorbing grape seed extract (GSE) SLNs (i.e., GSE/DA-SLNs) [12]. These last nanocarriers represent “multifunctional nanomedicines” that combine multiple biological functions into a single nanosystem with the capacity to achieve enhanced therapeutic responses, and the usefulness of this approach in brain diseases has been recently pointed out [13]. GSE/DA-SLNs were found to be not cytotoxic to both primary olfactory ensheathing cells and neuroblastoma SH-SY5Y cells by MTT test. Thus, the co-administration of these last SLNs can provide neurotransmitter DA useful for a “dopamine replacement strategy” and an antioxidant effect as GSE is able to reduce the ROS production, and, on the whole, this delivery system can constitute a novel approach for the treatment of PD. In this regard, it is useful to mention that polyphenols such as GSE show potent antioxidant activity against PD neurodegeneration, prevent dopaminergic neuronal cell loss and display reduction of oxidative stress and neuroinflammation [14].

From a formulation development point of view for in vivo studies, an approach could be to deliver SLNs into a final pharmaceutical form as a suspension of solid particles in an aqueous phase or in the form of freeze-dried powder leading to a nasal powder [15].

It is well known that to increase the physical stability of colloidal vehicles, including SLNs, freeze-drying still remains a good option. Freeze-drying technology mainly aims at obtaining a quick-re-dispersible powder from an aqueous liquid, therefore increasing either aqueous solubility or the shelf life of the active principles. Thus, an increase in the shelf-life of freeze-dried SLNs up to 1 year for both negatively- and positively-charged SLNs was reported [15]. However, when a freeze-drying cycle should be pursued, attention should be paid to the selection of the optimal cryoprotectant agent being necessary for a screening of different excipients at different concentrations in order to identify the most appropriate one. The following were employed for SLN protection: carbohydrates, such as glucose, lactose and sucrose; polyols, such as sorbitol; mannitol; and polymers, such as polyvinylpyrrolidone [15]. It is because such substances are able to replace water removal, occurring during the freeze-drying process, through hydrogen bonding formation with the polar groups of active drug substances involved in the freeze-drying process.

The present study was designed to test the feasibility of the nasal administration of GSE/DA-SLNs as re-dispersible dry powders obtained by freeze-drying as a novel approach for PD treatment. Thus, in order to obtain a final pharmaceutical form of GSE/DA-SLNs as a nasal powder for in vivo studies, we considered the option of preparing freeze-dried powders by lyophilization of a suspension of GSE/DA-SLNs by using sucrose or Methyl-β-cyclodextrin (Me-β-CD) as cryoprotectants of particular interest. While sucrose has already been used as an SLN cryoprotectant, as mentioned above, it seems that the use of Me-β-CD for SLN stabilization is unprecedented. However, it is well known that cyclodextrins (CDs), besides their application to increase the solubility and dissolution rate of poorly water-soluble drugs, as well as their capability to interact with biological membranes and to act as penetration enhancers of the nasal mucosa [16,17], can exert efficient cryoprotective effect towards colloidal carriers [18,19]. The choice of sucrose and Me-β-CD as cryoprotectants was made considering the different capabilities to permeate the biological membranes of these hydrophilic compounds due to their different molecular weight. The former hydrophilic cryoprotectant is regarded as “biological membrane-permeable” due to its low molecular weight, while the latter is regarded as “-non-permeable” due to its high molecular weight. Therefore, the aim of this study was to evaluate the effect of biological membrane permeable (sucrose) and non-permeable (Me-β-CD) cryoprotectants on the lyophilization of GSE/DA-SLNs. For this purpose, in this study, GSE/DA-SLNs were prepared following the whole freeze-drying cycle with or without cryoprotectants selected, and the resulting freeze-dried samples were characterized from physicochemical and physical stability points of view. In addition, solid-state studies on freeze-dried samples, including Raman spectroscopy, Scanning Electron Microscopy (SEM), atomic force microscopy (AFM) and X-ray Photoelectron Spectroscopy (XPS), were carried out. In vitro release studies and cell viability studies in the human nasal RPMI 2650 cell line were also performed. The obtained results were evaluated and discussed in light of the previously studied cryoprotectant-free GSE/DA- SLNs.

## 2. Results

### 2.1. Preparation and Physicochemical Properties of Cryoprotected SLNs

DA-co-GSE- and GSE-ads-DA-Gelucire^®^ 50/13-based SLNs were prepared following the melt homogenization method [20] and lyophilized by using sucrose or Me-β-CD as cryoprotectant agents, both used at the same concentration (6% *w*/*v*) as reported in Section 4.2. Table 1 deals with the main physicochemical properties of cryoprotected SLNs under investigation. From the reported data, it can be deduced that in the case of DA-co-GSE-SLNs, sucrose or Me-β-CD as a cryoprotectant agent did not significantly change the mean particle size before and after freeze-drying except for DA-co-GSE-SLNs-Me-β-CD (317 ± 61 nm vs. 614 ± 56 nm, *p* ≤ 0.05). Similarly, in the case of GSE-ads-DA-SLNs, it was noted that both cryoprotectants used did not modify in a significant manner the mean particle size before and after freeze-drying except for GSE-ads-DA-SLNs-sucrose (431 ± 99 nm vs. 1000 nm, *p* ≤ 0.01). On the other hand, the particle sizes of DA-co-GSE-SLNs and GSE-ads-DA-SLNs after freeze-drying with sucrose or Me-β-CD were bigger in a statistically significant way than the freshly prepared samples without cryoprotectant and freeze-drying cycle except for GSE-ads-DA-SLNs-Me-β-CD before lyophilization (i.e., 206 ± 94 vs. 287 ± 15) (Table 1).

The general increase in PDI values noted after freeze-drying cycles with both cryoprotectants may be indicative of a general broad and even plurimodal size distribution. This was confirmed by examining the distribution plots obtained by Photon Correlation Spectroscopy (PCS) (Figure 1a–d). On the other hand, a monomodal distribution was observed for freshly prepared samples without cryoprotectants and freeze-drying cycles (Figure 1e,f). The presence of microparticles in some samples may be due to aggregates of stuck particles in low amounts (and, hence, relatively small intensity of scattered light) [15].

In terms of zeta potential values, DA-co-GSE-SLNs-sucrose, both before and after freeze-drying, led to a significant modification of the external surface charge of DA-co-GSE-SLNs (−36.0 ± 0.8 nm and −29.4 ± 3.9 nm, respectively, *p* ≤ 0.01). In the case of DA-co-GSE-SLNs-Me-β-CD, a significant change in zeta potential value took place after freeze-drying (−6.9 ± 0.5 mV vs. −18.7 ± 2.4 mV, *p* ≤ 0.01). In the case of GSE-ads-DA-SLNs-sucrose, even though not statistically significant, a change in zeta potential after freeze-drying also occurs (−26.2 ± 3.9 mV vs. −20.6 ± 5.9 mV). When Me-β-CD was adopted, GSE-ads-DA-SLNs-Me-β-CD showed statistically significant changes in zeta potential values before and after lyophilization (−13.6 mV ± 1.3 vs. −10.3 ± 0.6 mV of GSE-ads-DA-SLNs-Me-β-CD before lyophilization and GSE-ads-DA-SLNs-Me-β-CD after lyophilization, respectively, *p* ≤ 0.05). Finally, it should be pointed out that the most negative surface potentials were noted in the case of lyophilized DA-co-GSE-SLNs- and GSE-ads-DA-SLNs-sucrose, suggesting an increased physical stability of these SLNs.

Concerning E.E.% in the SLNs, with respect to the cryoprotected DA-co-GSE-SLNs, DA content was equal to or less than that found in freshly prepared samples taken as control (Table 1). A similar trend for the E.E.% value of DA was also noticed for GSE-ads-DA-SLNs-Me-β-CD after freeze-drying in comparison to freshly prepared SLNs taken as control. Overall, GSE content was not markedly affected by the presence of sucrose or Me-β-CD in the SLNs both for DA-co-GSE-SLNs and for GSE-ads-DA-SLNs.

### 2.2. In Vitro DA or GSE Release from Cryoprotected SLNs in Simulated Nasal Fluid

In Figure 2a, the in vitro release profiles of the cryoprotected DA-co-GSE-SLNs and GSE-ads-DA-SLNs in Simulated Nasal Fluid (SNF) (pH 6.0 without enzymes) as receiving medium are reported. For all formulations, less than 10% of the intact DA was released into the medium, irrespective of the cryoprotectant agent used. Moreover, the highest DA release was noted from cryoprotected DA-co-GSE-SLNs-Me-β-CD followed by DA-co-GSE-SLNs-sucrose. Notably, GSE release took place only from DA-co-GSE-SLNs (Figure 2b), and the antioxidant agent was found in the SNF compartment in the range of 15–20%, either in the case of sucrose or Me-β-CD with an essentially identical kinetic profile. As seen, GSE was promptly released within two hours, showing a clear burst effect. In general, from the cryoprotected DA-co-GSE-SLNs and GSE-ads-DA-SLNs, both DA and GSE showed a biphasic release profile with an initial burst release followed by a slow release. Overall, throughout the 24 h of the in vitro release test, for all tested SLNs, no change of color was noticed in the SNF compartment during sampling, and, furthermore, no additional HPLC peak due to DA degradation was shown in the corresponding chromatograms. It is indicative that the neurotransmitter did not undergo any degradation, and this is probably due to the simultaneous presence of the antioxidant GSE [20].

### 2.3. Solid State Studies

#### 2.3.1. Physical Stability of Cryoprotected SLNs

Freeze-dried cryoprotected SLN pellets were stored at 4 °C, and the results of particle size monitoring for up to three months are shown in Figure 2c. Sucrose was seen to be a better cryoprotectant than Me-β-CD since the particle size of freeze-dried SLNs was retained (or did not change significantly) over time at 4 °C (Figure 2). On the other hand, for DA-co-GSE-SLNs-Me-β-CD, in the same storage conditions, mean diameters increased markedly (*** *p* < 0.001).

#### 2.3.2. Raman Spectroscopy

In order to investigate the possible interactions occurring between SLNs, with and without cryoprotectant in SNF, Raman spectra were recorded on the freeze-dried SLNs as such and after incubation in the SNF medium, as described in Section 4.7. The Raman spectra of the lipid Gelucire^®^, plain lyophilized SLNs, with or without cryoprotectant-SLNs, are shown in Figure 3 (traces a–h). In Table 2, the formulations were listed concerning the evaluation of two relevant intensities band ratios, which were previously adopted to study SLN formulations obtained from a different lipid and loading, as herein, the antioxidant mixture GSE [21].

In the spectral region between 2900 and 2800 cm^−1^, symmetric and antisymmetric C-H stretching signals of the -CH_2_ group were found, and, in particular, the I_2890_/I_2850_ ratio was calculated, being a marker for chain packing and conformational disorder. However, for DA-co-GSE-SLNs-sucrose and GSE-ads-DA-SLN-sucrose, no I_2890_/I_2850_ ratio was calculated due to the disappearance of the bands at 2850 cm^−1^ in the corresponding Raman spectra (Table 2) [22]. The bands at 1128 and 1062 cm^−1^ can be attributed, respectively, to the C-C symmetric and antisymmetric stretching vibrations, and it allowed for the calculation of the I_1115_/I_1050_ cm^−1^ ratio referring to the fluidity within the hydrocarbon chains (Table 2). In the Raman spectra of the SLNs, bands in the region between 800 and 850 cm^−1^ were found and ascribed to twisting and rocking vibrations [21,23]. In the lyophilized plain SLNs, DA-co-GSE-SLNs-sucrose and GSE-ads-DA-SLNs, the same Raman pattern of the lipid Gelucire was detected, and interestingly, no band intensity change was noticed. To prove sucrose’s presence as a cryoprotector, the peaks at 1130 and 1064 cm^−1^ can be attributed to the C-OH bending and C-O stretching, whereas the peak at 836 cm^−1^ is associated with the CH deformation round to the anomeric carbon atoms of the D-sucrose. Moreover, the spectral region before 800 cm^−1^ is associated with the vibrations of the skeleton of the sucrose molecule; in particular, the two bands at 744 and 800 cm^−1^ were linked to the ν(C-C) vibration of fructopyranose and fructofuranose, respectively. For the SLNs-Me-β-CD, the bands in the range of 3000 to 2900 cm^−1^ were associated with the -CH stretching of unsubstituted β-cyclodextrin, and in the range of 1440–1320 cm^−1^, the assignments were due to combinations of -C-C- stretching modes, -C-OH bond deformation and -CH_2_ group deformation of unsubstituted β-cyclodextrin [24,25].

When Raman spectra were acquired after SLN interactions with SNF (Figure 3b), the preincubation of the SLNs in such buffer did not lead to significant differences in the Raman patterns previously acquired on the SLN powders subjected to freeze-drying. With particular relevance to the Raman spectra concerning GSE-ads-DA-SLNs-sucrose compared with GSE-ads-DA-SLNs-sucrose after interaction with SNF (Figure 3b, d and c, respectively), in the Raman spectrum of Figure 3d, it is possible to detect a deformation of the bands in the ranges 800–940 cm^−1^, 1000–1200 cm^−1^, 1230–1350 cm^−1^ and 2800–3000 cm^−1^, leading to a wider and more flattened shape, probably due to the presence of the SNF.

#### 2.3.3. Morphology of the SLNs

SEM observations were carried out for DA-co-GSE-SLNs formulations selected as representative. SEM microphotographs of DA-co-GSE-SLNs and DA-co-GSE-SLNs-sucrose (both before and after freeze-drying using sucrose as a cryoprotectant) are shown in Figure 4. The morphology of both SLNs before freeze-drying appears to be very similar, namely, as folded sheets with rough surfaces. After the freeze-drying cycle, the roughness of the surface is decreased, particularly in the case where cryoprotectant is present. In any case, individual particles are not detectable by SEM.

Along with SEM observations, we have also performed morphological characterization of DA-co-GSE-SLNs and GSE-ads-DA-SLNs by using AFM before freeze-drying (Figure 5), providing crucial insights into their structural features. The observed round-shaped particles with even and smooth surfaces are indicative of a well-defined nano-scaled drug delivery system for DA and GSE. We found that the sizes of SLNs in the presence of sucrose or Me-β-CD were in close agreement with average particle sizes measured with PCS (Table 1). For instance, the average size of GSE-ads-DA-SLNs-Me-β-CD determined with AFM yielded an average size of 218 ± 71 nm, whereas the PCS measurement indicated a size of 206 ± 94 nm. A more noticeable difference was found for DA-co-GSE-SLNs-sucrose, where the AFM-derived particle size was measured as 397 ± 49 nm compared to an average size by PCS of 469 ± 40 nm. This formulation shows a relatively high PDI of 0.70, suggesting multiple particle populations with a high potential for particle aggregation.

After freeze-drying (Figure 6), we observed an increase in SLN sizes in the presence of sucrose or Me-β-CD while the original round-shaped structure was maintained. The particle size distribution of DA-co-GSE-SLNs-sucrose measured via AFM fitted well with its PCS intensity plot (Figure 1), showing multiple particle populations within the range of 100 and 3500 nm. Moreover, the integrity of DA-co-GSE-SLNs-sucrose after freeze-drying is more evident in the AFM phase image, indicating that SLNs are embedded in a cryoprotectant layer, a feature also observable in the SEM images (Figure 4). Similar results were obtained for other SLNs in the presence of Me-β-CD.

#### 2.3.4. X-ray Photoelectron Spectroscopy Analysis (XPS)

XPS analysis of the developed formulations was carried out in order to gain information on the surface presence of the neurotransmitter and—in the cases of the cryoprotected SLNs—of the cryoprotectant. The surface atomic composition of all the formulations, with or without sucrose (or Me-β-CD), was reported in Table 3. The DA presence on the surface can be related to the N1s detection. In this respect, no DA on the surface was detected only on DA-co-GSE-SLNs-Me-β-CD and GSE-ads-DA-SLNs-sucrose systems. Another interesting feature can be evidenced by C1s high-resolution spectra curve fittings of the cryoprotected formulations, reported in Figure 7. Indeed, it can be observed that, only for DA-co-GSE-SLNs-Me-β-CD and GSE-ads-DA-SLNs-sucrose systems, a C1s deconvolution was typical of the carbohydrate-based systems, thus evidencing a surface presence of sucrose (or Me-β-CD). In all the other systems, C1s curve-fitting was typical of plain SLNs (whose curve-fitting has already been reported by us [12]), thus evidencing a negligible presence of the cryoprotectant on the surface. Similar C1s curve-fittings were observed for the non-cryoprotected systems reported in Appendix A. Interestingly, when sucrose (or Me-β-CD) was not detected on the surface (because it was not allocated on the surface or completely absent in the formulation), then a DA surfacing was observed.

### 2.4. Effect of the Different DA- and GSE-Based Formulations on Cell Viability of Human Nasal Epithelial Cells

To evaluate the cyto-compatibility of the pure substances DA and GSE as well as of DA-co-GSE-SLNs and GSE-ads-DA-SLNs (these last lyophilized with and without cryoprotectant agents) with RPMI 2650 cells (i.e., a human cell line exhibiting epithelial morphology from the nasal septum), the viability of the mentioned cells when treated with such substances was studied. Figure 8 shows the results of a set of the viability assay according to Resazurin test performed with RPMI 2650 cells when treated with different concentrations of both pure DA and GSE at different concentrations tested, ranging from 12.5 to 100 µM for DA and from 2 to 100 µg/mL for GSE. DA did not decrease cell viability at any of the concentrations used, whereas GSE was found to be toxic over the concentration of 25 µg/mL.

In Figure 9a and Figure 10a, the results of a 6 h incubation treatment of RPMI 2650 cells with DA-co-GSE-SLNs and GSE-ads-DA-SLNs are depicted, respectively. In the case of Figure 10a, we found a statistically significant increase in cell viability values up to 15% after treatment with the control formulation GSE-ads-DA-SLNs. Figure 9b,c and Figure 10b,c show the effects on cell viability of RPMI 2650 cells treated with the cryoprotected DA-co-GSE-SLNs and GSE-ads-DA-SLNs at different time points. Statistical analysis using ANOVA and Dunnett post hoc tests showed that there were no statistically significant differences between different treatment conditions, at different dilutions, and at different times. Cell viability of RPMI 2650 cells treated with DA-co-GSE-SLNs-Me-β- CD and GSE-ads-DA-SLNs-Me-β- CD did not show any statistically significant difference (according to ANOVA with Dunnett’s post hoc test) when the two types of SLNs were tested at different concentrations (Figure 9b,c and Figure 10b,c). By performing the statistical Sidak test to evaluate the differences between the SLNs with and without cryoprotectants, we found that the presence of cryoprotectants did not significantly affect RPMI cell viability with respect to the incubation of the cells with the cryoprotectant-free SLNs at any of the tested time points. In the case of SLNs with and without cryoprotectants, we found no significant difference occurring in 6 h (Figure 9a and Figure 10a), whereas a slight reduction in RPMI 2650 cell viability occurred at 12 and 24 h (Figure 9b,c and Figure 10b,c).

## 3. Discussion

The aim of the present work was to evaluate the effect of a small molecule (biological membrane-permeable, sucrose) and of an oligomer (biological membrane non-permeable, Me-β-CD) cryoprotectant on the lyophilization of a suspension of GSE/DA-SLNs. It is in order to deliver the resulting solid nanocarriers as powders by intranasal route as a new approach for PD treatment. As shown in Table 1, both in the case of DA-co-GSE-SLNs- and GSE-ads-DA-SLNs-cryoprotectant, in general, particle size differences before and after freeze-drying were not statistically significant. In contrast, the particle size of DA-co-GSE-SLNs and GSE-ads-DA-SLNs after freeze-drying with both cryoprotectants used was bigger in a statistically significant way than the freshly prepared samples without cryoprotectant and the freeze-drying process. This last outcome is consistent with the general increase in PDI values noted after freeze-drying cycles with both cryoprotectants, accounting for a general broad and even plurimodal distribution, as shown in Figure 1 [15]. As for the huge size difference between the freshly prepared nanosuspensions DA-co-GSE-SLNs or GSE-ads-DA-SLNs (lines 1 or 8 in Table 1) and the corresponding ones before freeze-drying cycle (lines 2 and 9 in Table 1), it may be due to physical instability of the corresponding nanoparticles during the two weeks of storage at 4 °C in the refrigerator. All these results are in agreement with that reported in the literature for other negatively charged SLNs [15,26,27]. An interesting outcome of the physicochemical characterization of the nanocarriers herein studied refers to the most negative surface potentials observed by us in the case of lyophilized DA-co-GSE-SLNs- and GSE-ads-DA-SLNs-sucrose suggesting an increased physical stability of these cryoprotected SLNs. The zeta potential changed from −29.4 mV to −20.6 ± 5.9 mV (zeta potential values that still result in an adequate range for sufficient stability), evidencing the sufficient colloidal stability of these SLNs. This suggestion is confirmed by the results of the physical stability study that evidenced the positive role played by sucrose as a cryoprotectant, which considerably affects the stability of the SLNs studied (Figure 2c).

To justify the results of the in vitro release kinetic study of DA and GSE shown in Figure 2a,b, several factors should be taken into account, including the results of XPS analysis. Thus, the highest neurotransmitter release from DA-co-GSE-SLNs-Me-β-CD, for which DA on the surface was not detected by XPS, may be accounted for the increase in surface roughness with a higher surface area of the freeze-dried powder due to the presence of Me-β-CD in which, as shown in the AFM phase image, such SLNs are embedded in a cryoprotectant layer [28]. Appendix A displays the topographic profile derived from the height image of GSE-ads-DA-SLNs in the presence of sucrose. The profile exhibited a relatively smooth surface with no discernible elevations or depths, indicating minimal porosity on the SLNs’ surface. In contrast, Appendix A presents the topographic profile obtained from the height image of DA-co-GSE-SLNs in the presence of Me- β-CD. The profile displayed an uneven surface with pores characterized by diameters of 2–15 nm, suggesting the presence of higher surface roughness on the particle’s surface. Indeed, the increase of nanocarrier surface roughness arising from lyophilization, also detected by SEM microphotographs, facilitates the release of the active substance, and CDs are useful excipients in the production of large porous particles (Appendix A, [29]). Indeed, the SEM shows the surface morphology of the composite in as-prepared conditions, whereas the AFM investigated that of the individual SLNs after isolating them through the ad hoc procedure devised and described in our paper. To be sure, further specific experiments are necessary to draw definitive conclusions about the presence and diameters of pores in DA-co-GSE-SLNs-Me-β-CD SLNs. The favorable effect of the porosity on the release kinetic may also explain the higher neurotransmitter release from DA-co-GSE-SLNs-Me-β-CD compared with that of DA-co-GSE-SLNs-sucrose for which neurotransmitter was detected on the nanocarrier surface (Figure 2a). The gradual release of DA observed after the initial burst effect may be due to the erosion of the lipid matrix of the SLNs [27]. The reduced DA amount released from GSE-ads-DA-SLNs-Me-β-CD and from GSE-ads-DA-SLNs-sucrose (Figure 2a) may be due to the shielding effect exerted by the surface adsorbed antioxidant agent GSE on the SLNs. Concerning the GSE release kinetic observed only from cryoprotected DA-co-GSE-SLNs-Me-β-CD or DA-co-GSE-SLNs-sucrose (Figure 2b), at present, this result is difficult to account for. Our hypothesis is that such release does not occur from cryoprotected GSE-ads-DA-SLNs because the -OH groups of the antioxidant agent GSE are involved in hydrogen bonding interactions with PEG residues of the lipid Gelucire^®^. Such interactions do not occur when DA and GSE are co-encapsulated in SLNs with cryoprotectants. Furthermore, the same release profile observed for DA-co-GSE-SLNs-Me-β-CD and DA-co-GSE-SLNs-sucrose is not surprising since the same amount of GSE is employed to prepare both these PEGylated SLNs.

As for the investigation aimed to evidence possible interactions with nasal fluids of cryoprotected SLNs, we have recently reported that a useful approach is based on the use of Raman spectroscopy [21], herein applied on the freeze-dried SLNs as such and after incubation in the SNF medium. In particular, as mentioned above, it has been shown that the I_2890_/I_2850_ ratio is an indication of chain packing and conformational disorder, while the I_1115_/I_1050_ cm^−1^ ratio refers to the fluidity within the hydrocarbon chains [21]. In the case herein examined, interaction with SNF is based on band deformation observed in the ranges 800–940 cm^−1^, 1000–1200 cm^−1^, 1230–1350 cm−^1^and 2800–3000 cm^−1^, leading to a flattened shape, probably due to the presence of the SNF. Thus, in perspective, Raman spectroscopy confirms its interesting potential when applied to provide more detailed information on these systems.

On the other hand, XPS provided important information on the surface presence of the neurotransmitter as well as even the cryoprotectant when it occurs. Indeed, XPS analysis showed in DA-co-GSE-SLNs-sucrose after freeze-drying a negligible sucrose presence and detection of DA on the surface (Table 3). A similar situation could also be described for SLN structures, but at the same time, it could not be ruled out that a competition for surface localization could take place between DA and sucrose (or Me-β-CD). In this case, it should be supposed that the higher density of the -OH groups of sucrose (or Me-β-CD) interacting with PEG residues of the lipid Gelucire^®^ can displace DA from the external surface of the SLNs, thus determining DA leakage. Overall, such a reduction in DA levels did not cause the absence of the neurotransmitter delivered in SNF, as shown in Figure 2. Indeed, XPS analysis evidenced a surface presence of the cryoprotectant only on DA-co-GSE-SLNs- Me-β-CD and GSE-ads-DA-SLNs-sucrose, as shown in Figure 7. Therefore, although the presence on the surface of some of the developed formulations of an additional excipient, such as sucrose (or Me-β-CD), potentially shields/limits the release, DA was delivered in the nasal compartment, leading to the reasonable prediction that in vivo the neurotransmitter could follow nose-to-brain-pathway.

In the development of intranasal drug delivery systems for brain targeting, it is necessary to assess its cytotoxicity through the nasal epithelium. Concerning the biological evaluation of the investigated SLNs, the RPMI 2650 cell line was herein adopted as they represent a cell model widely explored for mimicking human nasal epithelium [30,31,32,33]. When RPMI 2650 cells were exposed to the cryoprotectant free-SLN formulations, no toxic effect appeared at the different concentrations and time points tested (Figure 9 and Figure 10). Moreover, full cell viability was found when RPMI 2650 was exposed at different time points to SLNs-sucrose (Figure 9 and Figure 10). Similarly, the addition of Me-β-CD as a cryoprotectant to both DA-co-GSE-SLNs and GSE-ads-DA-SLNs determined only a very slight, not significant reduction in cell viability in comparison to the absence of Me-β-CD, irrespectively of the time of treatment (Figure 9b,c and Figure 10b,c). To account for this very low tendency reduction in cell viability observed when Me-β-CD was used as a cryoprotectant, the mentioned biological membrane non-permeable feature of this methylated CD could be invoked, and its well-known rapid extraction capacity of cholesterol from cell membrane [34]. Hence, at the longest incubation time points, it could be expected that the cryoprotectant Me-β-CD could damage the cell membrane as a consequence of cholesterol complexation in the cavity of the host Me-β-CD. It seems that sucrose works better as a cryoprotectant than Me-β-CD for both types of SLN formulations, and for further development of the nasal powders containing SLNs, it would be taken into account.

Regarding the effect of the GSE effect on the cells, we observed a statistically significant decrease in RPMI 2650 viability at concentrations higher than 25 µg/mL at any incubation time point (Figure 8, Figure 9 and Figure 10), confirming Lin et al.’s findings [35]. The authors have tested a GSE range of concentrations from 25 to 100 μg/mL for 24 h in HL-60/ADR human acute myeloid leukemia cell line, revealing a statistically significant reduction of viability already starting from 50 μg/mL and a reduction of viability from 20% to about 60% after treatment with 50–100 μg/mL. However, differences due to the cell line employed and the sensitivity to the antioxidant effect are to be taken into account. In fact, Junior et al. [36] tested different GSE concentrations, in particular 50, 250, 500 and 1000 µg/mL, on cancer cell lines such as A549 (lung carcinoma cell line), HepG2 (hepatocellular carcinoma cell line), and Caco-2 (colorectal adenocarcinoma cell line) and healthy cell lines such as IMR90 (fibroblasts from normal lung tissue) for 48 h. Surprisingly, the GSE extracts did not show any cytotoxic or antiproliferative effect on these cell lines, and in many cases, there was even an increase in cell viability of more than 200%. Altogether, for PD application, the delicate balance between the exogenous supply of antioxidant species (like GSE polyphenols mediating neuroprotective actions in serotonergic transmission in PD) and the need to preserve inside the cells some radical oxygen species requires further investigation. Indeed, in PD, the so-called “mitochondrial disfunction” markedly contributes to the clinical aspects of the pathology, together with the reduction of DA levels in the Substantia Nigra. Hence, the combination of anti-PD and antioxidant agents seems promising for clinical applications [37,38,39].

## 4. Materials and Methods

### 4.1. Materials

Grape seed extract containing ≥ 95.0% proanthocyanidins was received as a gift by Farmalabor (Canosa di Puglia, Italy). Gelucire^®^ 50/13 was kindly donated by Gattefossé (Milan, Italy). Dopamine hydrochloride, sucrose, carboxyl ester hydrolase (E.C. 3.1.1.1, 15 units/mg powder), Tween^®^ 85, as well as the salts used for buffer preparation, were bought from Sigma-Aldrich (Milan, Italy). Methyl-β-cyclodextrin (Me-β-CD, Mw 1320 Da, average substitution degree 1.8) was a gift from Wacker Chemie (Milan, Italy) and kept in a desiccator until use. For cell cultures, advanced Minimum Essential medium (A-MEM) was purchased from Gibco-Thermo Fisher Scientific (Waltham, MA, USA). Advanced Minimum Essential medium (A-MEM) was purchased from Gibco-Thermo Fisher Scientific (Waltham, MA, USA). Heat-inactivated fetal bovine serum (FBS) was purchased from Euroclone S.p.A (Pero, Italy). GutaMAX™ Supplement was acquired from Biowest (Nuaillé, France). Trypsin EDTA 0.25% was purchased from Elabscience (Huston, TX, USA). Alamar Blue and Resazurin were purchased from Bio-Rad (Hercules, CA, USA) and Biotium (Fremont, CA, USA), respectively. In this work, double distilled water was used, and all other chemicals were of reagent grade.

### 4.2. Solid Lipid Nanoparticle Preparation

DA-loaded Gelucire^®^ 50/13 SLNs were prepared following the melt homogenization method. Briefly, Gelucire^®^ 50/13 (60 mg) was melted at 70 °C. GSE (6 mg) were dispersed in the aqueous phase made of surfactant (Tween^®^ 85, 60 mg) and 1.37 mL diluted acetic acid (0.01%, *w*/*v*) in a separate vial under homogenization at 12,300 rpm with an Ultra-Turrax model T25 apparatus (Janke and Kunkel, IKA^®^-Werke GmbH & Co., Staufen, Germany) and left to equilibrate for 30 min at 70 °C. Then, 10 mg of DA was added to the aqueous phase, the resulting mixture was emulsified with the melted Gelucire^®^ 50/13, and the emulsion was homogenized at 12,300 rpm for 2 min by Ultra-Turrax system. Then, the nanosuspension was cooled at room temperature and allowed to achieve DA co-encapsulating GSE SLNs. Such SLNs were centrifuged (16,000× *g*, 45 min, Eppendorf 5415D, Hamburg, Germany), the pellet was harvested and re-suspended in distilled water for further studies, and the supernatant was discarded. Throughout the manuscript, the resulting SLNs were abbreviated “DA-co-GSE-SLNs”. For GSE-adsorbing DA SLNs, DA-loaded SLNs were first formulated as reported elsewhere [10,12] but starting from 20 mg of DA rather than 10 mg to force neurotransmitter initial cargo. After cooling the particles down at room temperature, an aliquot of 0.5 mL of the resulting DA-SLNs was incubated with 1 mL of GSE aqueous solution (1 mg/mL concentration) at room temperature for 3 h under light protection and maintaining mild stirring (50 oscillations/min). Then, the mixture was centrifuged at 16,000× *g* for 45 min (Eppendorf 5415D), and the pellet was re-dispersed in distilled water, whereas the supernatant was discarded. Throughout the manuscript, the resulting SLNs were abbreviated as “GSE-ads-DA-SLNs”.

For both DA-co-GSE-SLNs and GSE-ads-DA-SLN, the effect of two cryoprotectant excipients was studied following hints in the literature [14]. After preparing a solution of sucrose or Me-β-CD at the concentration of 60 mg/mL in double distilled water, 1 mL of each solution was added to the re-suspended pellet of DA-co-GSE-SLNs and GSE-ads-DA-SLN prior to freeze-drying cycle (72 h at T = −50 °C; *p* = 0.1 mbar; Lio Pascal 5 P, Milan, Italy). Below, the resulting SLNs were abbreviated “DA-co-GSE-SLNs-sucrose (or -Me-β-CD)” and “GSE-ads-DA-SLNs-sucrose (or -Me-β-CD)”.

### 4.3. Quantification of DA and GSE

The quantitative determinations of DA and GSE were performed by HPLC according to procedures previously reported with slight modifications [40,41]. Briefly, the HPLC apparatus included a Waters Model 600 pump (Waters Corp., Milford, MA, USA), a Waters 2996 photodiode array detector and a 20 μL loop injection autosampler (Waters 717 plus). A Synergy Hydro-RP (25 cm × 4.6 mm, 4 μm particles; Phenomenex, Torrance, CA, USA) was the stationary phase, and a 0.02 M potassium phosphate buffer (pH 2.8: CH_3_OH 70:30 (*v*:*v*)) was adopted as mobile phase. The isocratic mode was selected for column elution at the flow rate of 0.7 mL/min, and, under such chromatographic conditions, the retention times of DA and GSE were found to be equal to 5.5 min and 12 min (see Appendix A), respectively.

To determine DA and GSE content, herein, quantification of the active principles occurred after freeze-drying of the different types of cryoprotected SLNs, which underwent enzymatic digestion by esterases [19]. The enzyme was dissolved at 12 I.U./mL in phosphate buffer (pH 5), and 1–2 mg of freeze-dried SLNs were incubated with 1 mL of the enzyme solution for 30 min in an agitated (40 rpm/min) water bath set at 37 °C (Julabo, Milan, Italy). Then, the resulting mixture was centrifuged (16,000× *g*, 45 min, Eppendorf 5415D), and the obtained supernatant was subjected to HPLC analysis, as above. The encapsulation efficiency (E.E.%) was calculated by Equation (1):E.E.% = DA (GSE) in the supernatant after esterase assay/Total DA (GSE) × 100(1)
where total DA (GSE) is intended as the starting amount of each substance used for SLN preparation.

Furthermore, drug-loading (D.L.) was also calculated by Equation (2):D.L.% = DA (GSE) in the supernatant after esterase assay/Total Gelucire^®^ 50/13 adopted in the formulation × 100 (2)

This study was performed in triplicate.

### 4.4. Physicochemical Characterization of SLNs

Mean particle size and polydispersity index (PDI) of the SLNs in the presence and in the absence of cryoprotectant were determined according to Photon Correlation Spectroscopy (PCS) of the ZetasizerNanZS (ZEN 3600, Malvern, UK) apparatus. Samples of SLNs were analyzed in terms of particle size, PDI and zeta potential before and after freeze-drying. They were introduced in folded capillary zeta cells with a measurement angle of 173° Backscatter. Particle size and PDI values of the SLNs were measured at 25 °C by re-dispersing each of them in 1 mL of double distilled water, followed by a brief sonication. Then, the nanosuspensions underwent a further dilution in double distilled water (1:200, *v*/*v*) prior to being analyzed for particle size. For ζ-potential measurements, laser Doppler anemometry technique was adopted (ZetasizerNanoZS, ZEN 3600, Malvern, UK) by using the same dilution as described above for size analysis.

### 4.5. Physical Stability of SLNs with Cryoprotecant Agents

To evaluate GSE-ads-DA-SLNs and DA-co-GSE-SLNs’ physical stability in the presence of sucrose and/or Me-β-CD as cryoprotectant agents, SLN pellets were subjected to lyophilization upon the conditions described in Section 4.2. Then, solid pellets were stored at 4 °C in the refrigerator for up to 12 weeks. At different time points, SLNs were reconstituted by adding 1 mL of double distilled water under gentle vortexing and particle size was acquired according to the mean diameter analysis procedure described in Section 4.4. For each type of SLN, the assay was performed in triplicate.

### 4.6. In Vitro DA and GSE Release from Cryoprotected SLNs in SNF

The DA and GSE release in SNF [42] (pH 6.0, without enzymes) from freeze-dried cryoprotected SLNs (i.e., DA-co-GSE-SLNs-sucrose or DA-co-GSE-SLNs-Me-β-CD, GSE-ads-DA-SLNs-sucrose and GSE-ads-DA-SLNs-Me-β-CD) was performed as follows. Firstly, an amount of lyophilized SLNs corresponding to 7–8 mg of DA and 5–6 mg of GSE were weighted. The receiving medium was represented by 20 mL of SNF thermostated at 37 ± 0.1 °C in an agitating (40 rpm/min) water bath (Julabo, Milan, Italy). At scheduled time points (0–1–3–6–24 h), 0.8 mL of the receiving SNF was withdrawn and replaced with 0.8 mL of fresh SNF. Afterward, each sample was centrifuged at 16,000× *g* for 45 min (Eppendorf 5415D, Germany), and the amounts of the neurotransmitter and the antioxidant GSE delivered were quantified in the resulting supernatants by HPLC, as described above, and plotted against the time. All the release experiments in SNF were carried out in triplicate.

### 4.7. Raman Spectroscopy

After freeze-drying SLNs in the presence and in the absence of cryoprotectants, Raman spectra were recorded using an InVia Renishaw microscope (Renishaw, Wotton-under-Edge, UK) equipped with a 532 nm and 633 nm laser in the presence of a 100× objective. The specimens were placed onto a microscope slide, and the range examined was 400–3500 cm^−1^ using a 600 (L)/mm grating. The parameters used for Raman spectra acquisitions were 2 s exposure, 20 accumulations, 100% of laser power and 60–300 s bleaching. In order to gain insight into the interactions occurring between SLNs (provided or not of cryoprotectant) and the nasal compartment, freeze-dried SLNs were incubated in 20 mL of SNF at 37 °C for 3 h in an ISCO thermostated incubator (Isco, Trieste, Italy) without any agitation. Once the incubation was over, a few drops of the resulting suspensions were placed on different slides and allowed to evaporate overnight prior to being subjected to Raman investigations.

### 4.8. SEM Observations

Energy-dispersive X-ray analysis (EDX) and Scanning Electron Microscopy (SEM) were performed using a VegaII microscope (Tescan, Brno, Czech Republic) with a Quantax elemental detector (Bruker, Billerica, MA, USA). The specimens were placed onto a microscope stub, and a sputter coater (Cressington, Watford, UK) was used to cover specimens with a thin gold layer. Then, the specimens were introduced into the SEM chamber and analyses were performed using a high voltage of 20 kV.

### 4.9. AFM Studies

Freshly cut silicon wafers (0.5 cm × 0.5 cm) were glued onto a microscopic glass slide and thoroughly cleaned with pure isopropanol and ultrapure water. Afterward, SLN suspensions were diluted 1:10 (*v*/*v*) with ultrapure water and pipetted onto the silicon wafers. Samples were then incubated for 30 min to allow adherence of particles to the wafer’s surface. Excess water was shaken off, and samples were mounted onto the stage of a NanoWizard^®^-3 NanoScience AFM (JPK/Bruker, Berlin, Germany), which was vibration-damped and located in an acoustic isolation chamber.

Samples were measured at RT in AC mode in air [43]. A target amplitude of 1 V was selected, and the relative setpoint was set to 90%. A commercial AFM cantilever HQ:NSC16/Al BS with a resonance frequency of 160 kHz and a force constant of 45 N/m was used for all measurements. Settings like gains, setpoint and drive amplitude were adjusted during the measurement to optimize image quality. Scan speed was set to 0.6 Hz and 1.5 Hz for scan sizes of 10 µm × 10 µm and 1.5 µm × 1.5 µm, respectively. To evaluate the particle diameters, 75 particles were evaluated (from height mode) for each sample. Raw images were then edited by using JPK data processing software., version 6.1.62). A polynomial fit was subtracted from each scan line independently and by using a limited data range. Small imaging errors were corrected by replacing lines by interpolating, or outliers were replaced by neighboring pixels.

### 4.10. XPS Investigations

XPS analyses were performed using a scanning microprobe PHI 5000 VersaProbe II purchased from Physical Electronics (Chanhassen, MN, USA). The instrument was equipped with a micro-focused monochromatized AlKα X-ray radiation source. Freeze-dried SLNs (with or without cryoprotectant) were examined in HP mode with an X-ray take-off angle of 45°, with an instrument base pressure of ~10^−9^ mbar. The scanned area sizes were 1400 × 200 μm. Wide scans and high-resolution spectra were recorded in FAT mode (pass-energy equal to 117.4 eV and 29.35 eV, respectively). For curve-fitting of the high-resolution spectra, the commercial MultiPak software version 9.9.0.8 was used. Adventitious carbon C1s was set as the reference charge (284.8 eV).

### 4.11. Cell Culture

The RPMI 2650 cells, sourced from the European Collection of Cell Cultures, Health Protection Agency, were donated by Dr. Katja Kristan (University of Ljubljana, Slovenia). They were cultured in 25 cm^2^ polystyrene tissue culture flasks using A-MEM. The medium was supplemented with 4 mM GlutaMAX™ and 2.5% FBS. The cell cultures were maintained at 37 °C in a >95% humidified atmosphere of 5% CO_2_ in air, with media changes on alternative days [29,30].

### 4.12. Cell Viability Assay

Resazurin protocol (Biotium, Fremont, CA, USA) was used to assess RPMI 2650 cell viability [44]. For these assays, 15,000 cells per well in a 96-well multi-well plate were grown, and 72 h after seeding, culture medium was removed and replaced with 200 µL of fresh medium containing SLNs, whose effects were evaluated after 6, 12 and 24 h according to resazurin viability assay. At the appropriate incubation times, the medium was removed from each well, 100 µL of ten-times-diluted resazurin in PBS was added, and the plate was incubated for approximately 90 min until the control changed from blue to pink. Fluorescence was then evaluated using a FLUOstar^®^ Omega multi-plate reader (BMG Labtech, Ortenberg, Germany) at 560 nm (excitation) and 590 nm (emission). Cryoprotected SLNs based on the concentration of DA were diluted using A-MEM in order to obtain the following DA concentrations: 100, 75, 50 and 25 µM. Once DA concentration was fixed in the SLNs, then the GSE range of concentrations was found to be between 23.8 µg/mL and 0.02 ng/mL. Moreover, for SLNs cryoprotected by sucrose and Me-β-CD, 0.8–0.1 mg/mL and from 1 mg/mL to 0.078 mg/mL were the ranges examined for each excipient, respectively. Each formulation was tested two to three times in triplicate.

### 4.13. Statistical Analysis

For physicochemical data, mean ± SD and statistical evaluation were obtained by Prism v. 5.0 (GraphPad Software Inc., La Jolla, CA, USA). Multiple comparisons were based on one-way analysis of variance (ANOVA), with Bonferroni’s post hoc test, and differences were considered significant when *p* < 0.05. Biological data are presented as means ± standard error (s.e.m.). The effect of treatments was analyzed using GraphPad Prism 9.0.0 software for Windows (GraphPad Software, San Diego, CA, USA, www.graphpad.com, accessed on 15 April 2021) by ANOVA. The Dunnett post hoc test and the Sidak multiple comparison test were performed. Particularly, the Dunnett test compared the means of several experimental groups with the mean of a control group to determine whether there was a difference; the Sidak test was used to compare viability values regarding the same concentration of samples with and without cryoprotectants. Differences were considered statistically significant at *p* < 0.05.

## 5. Conclusions

Herein, SLNs loaded with two hydrophilic active principles, namely DA and GSE, were formulated in the presence of sucrose or Me-β-CD as cryoprotectant agents in view of the development of a nasal solid dosage form for PD application. Particle size retention at 4 °C together with RPMI 2650 cell viability evaluation evidenced some benefits to adopt sucrose better than Me-β-CD. Thus, the physical stability study evidenced the positive role played by sucrose as a cryoprotectant, which considerably affects the stability of the SLNs studied, as proved by particle size retention up to three months of storage at 4 °C. Overall, considering the improvement of SLN stability, the whole freeze-drying cycle using sucrose as a cryoprotectant is advantageous in this case. In addition, it should also be taken into account that RPMI 2650 full viability was retained up to 24 h of incubation via the use of sucrose rather than Me-β-CD as a cryoprotectant of SLNs. Finally, the outcomes of this research work will constitute a platform for further studies addressed to obtain the aerosolization of the nasal powders herein described and build the device containing such pharmaceutical dosage forms.

## Figures and Tables

**Figure 1 molecules-28-07706-f001:**
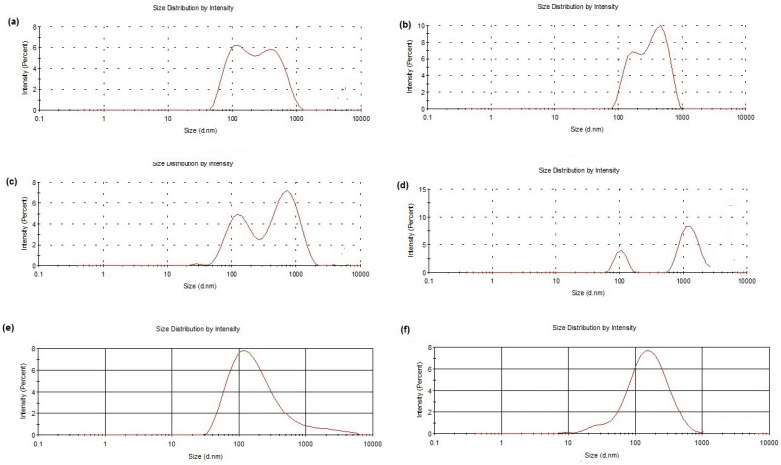
Size distribution plots of DA-co-GSE-SLNs-sucrose after freeze-drying (a. f. d.) (**a**), GSE-ads-DA-SLNs-sucrose a. f. d. (**b**), DA-co-GSE-SLNs-Me-β-CD a. f. d. (**c**), GSE-ads-DA-SLNs-Me-β-CD a. f. d. (**d**), DA-co-GSE-SLNs freshly prepared without any cryoprotectant and freeze-drying cycle (**e**) and GSE-ads-DA-SLNs freshly prepared without any cryoprotectant and freeze-drying cycle (**f**) are shown.

**Figure 2 molecules-28-07706-f002:**
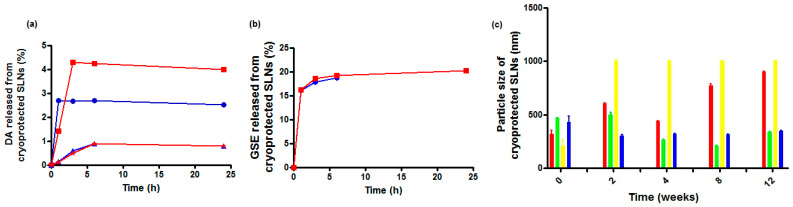
Panel (**a**): in vitro release profiles of DA in SNF from cryoprotected DA-co-GSE-SLNs-Me-β-CD (red-filled square and line), from DA-co-GSE-SLNs-sucrose (blue-filled circles and line), from GSE-ads-DA-SLNs-Me-β-CD (red-filled stars and line) and from GSE-ads-DA-SLNs-sucrose (blue-filled triangles and line). Panel (**b**): in vitro release profiles of GSE in SNF from cryoprotected DA-co-GSE-SLNs-Me-β-CD (red-filled square and line) and from DA-co-GSE-SLNs-sucrose (blue-filled circles and line). Standard deviations measured for the data points in (**a**,**b**) were very small and even smaller than the size of the markers in the plots. Panel (**c**): particle size changes of cryoprotected SLNs at 4 °C over three months of storage after freeze-drying. DA-co-GSE-SLNs-Me-β-CD (red bars); DA-co-GSE-SLNs-sucrose (green bars); GSE-ads-DA-SLNs-Me-β-CD (yellow bars); GSE-ads-DA-SLNs-sucrose (blue bars).

**Figure 3 molecules-28-07706-f003:**
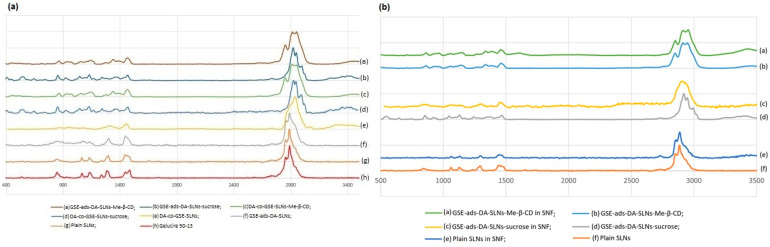
(**a**) Raman spectra of different SLN formulations after freeze-drying; (**b**) Raman spectra of different SLN formulations that were freeze-dried after incubation in SNF (See Section 4.7).

**Figure 4 molecules-28-07706-f004:**
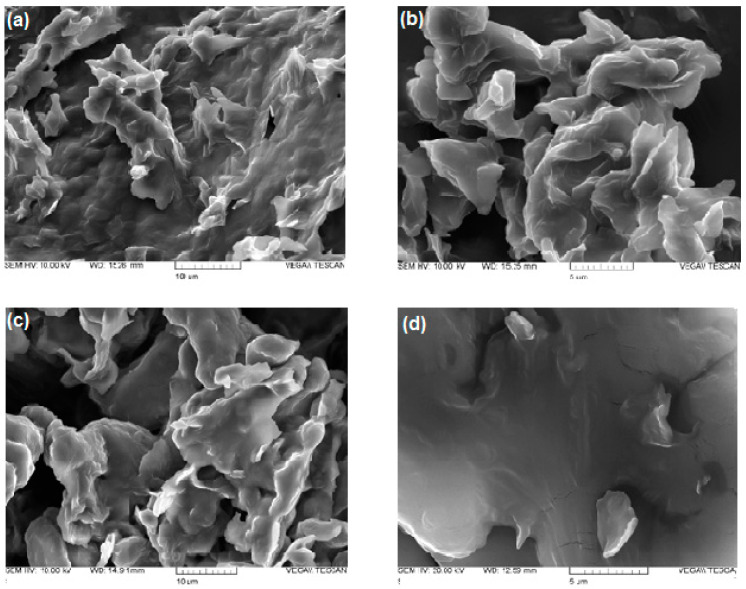
SEM pictures of DA-co-GSE-SLNs before (panel (**a**)) and after freeze-drying (panel (**b**)); SEM pictures of DA-co-GSE-SLNs-sucrose before (panel (**c**)) and after freeze-drying (panel (**d**)).

**Figure 5 molecules-28-07706-f005:**
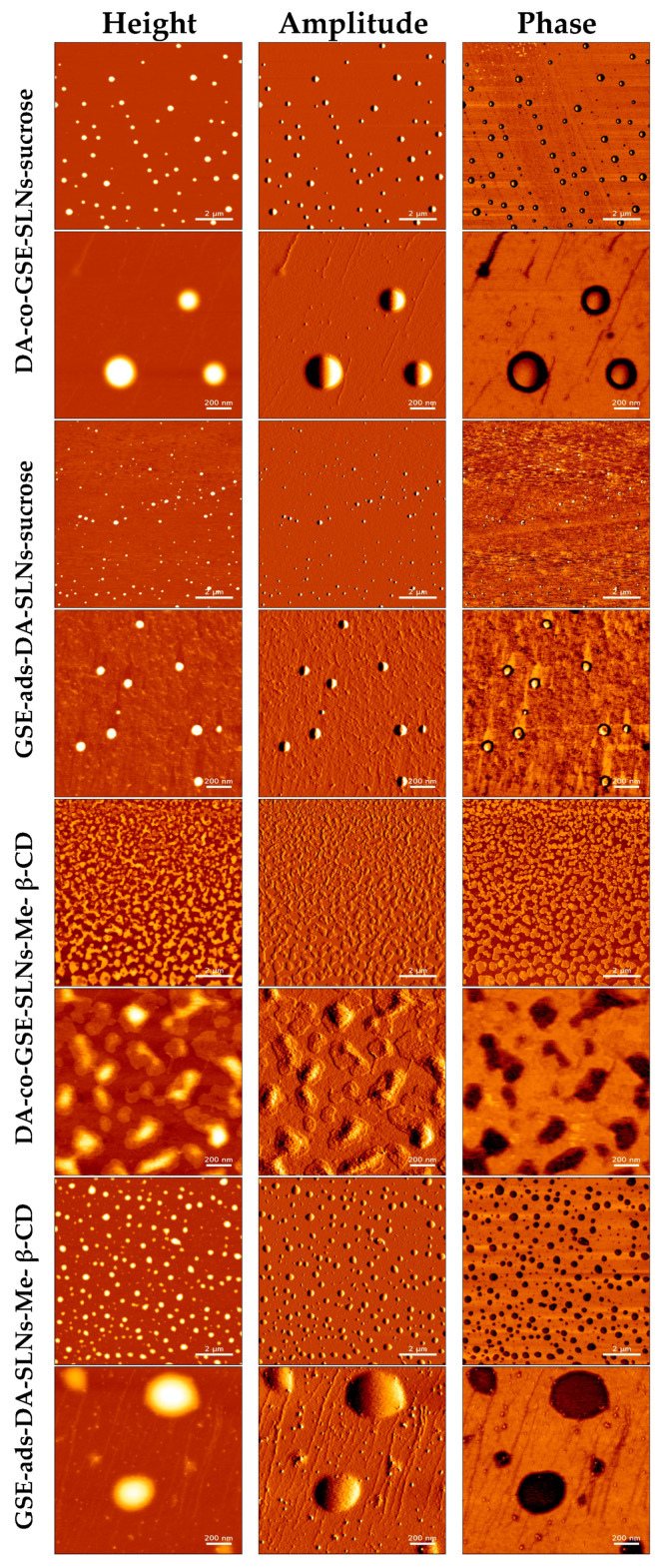
Visualization of various GSE/DA SLNs with sucrose or Me-β-CD as a cryoprotectant by atomic force microscopy before freeze-drying.

**Figure 6 molecules-28-07706-f006:**
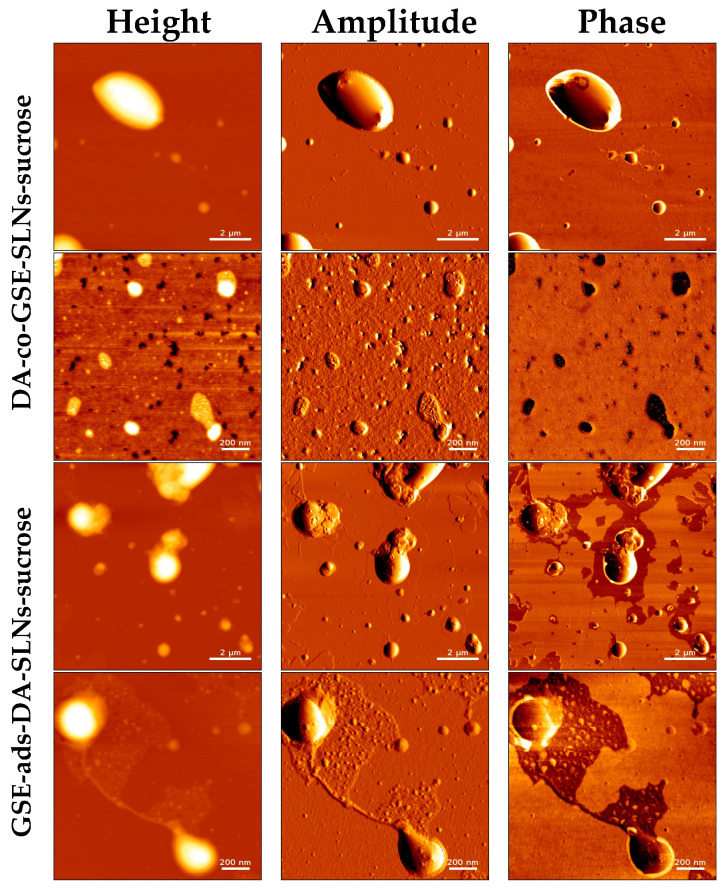
Visualization of GSE/DA SLNs-sucrose by atomic force microscopy after freeze-drying.

**Figure 7 molecules-28-07706-f007:**
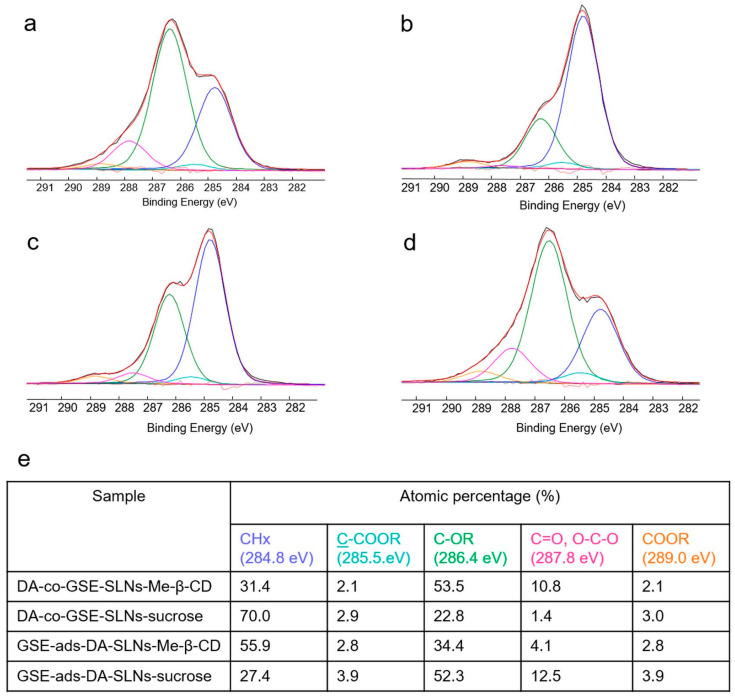
Curve-fittings (red lines) of C1s signals (black lines) relevant to DA-co-GSE-SLNs-Me-β-CD (**a**), DA-co-GSE-SLNs-sucrose (**b**), GSE-ads-DA-SLNs-Me-β-CD (**c**), GSE-ads-DA-SLNs-sucrose (**d**) and the peak attributions, binding energies and atomic percentages of the relevant samples (**e**). Uncertainty on BE peak positions was ±0.2 eV.

**Figure 8 molecules-28-07706-f008:**
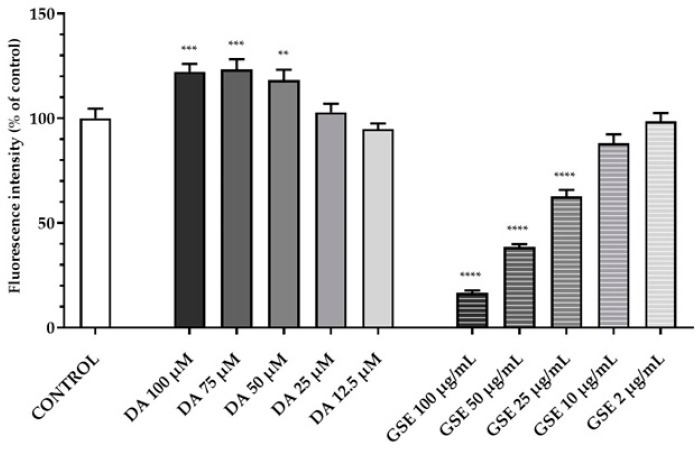
Resazurin viability assay on RPMI 2650 cells after 24 h treatment with pure substances DA in the range of 12.5–100 µM and GSE in the range of 2–100 µg/mL. Statistical analysis using ANOVA and Dunnett post hoc test: 6 < n < 9 for all the treatments; ** *p* < 0.01; *** *p* < 0.001; **** *p* < 0.0001. Controls were untreated cells.

**Figure 9 molecules-28-07706-f009:**
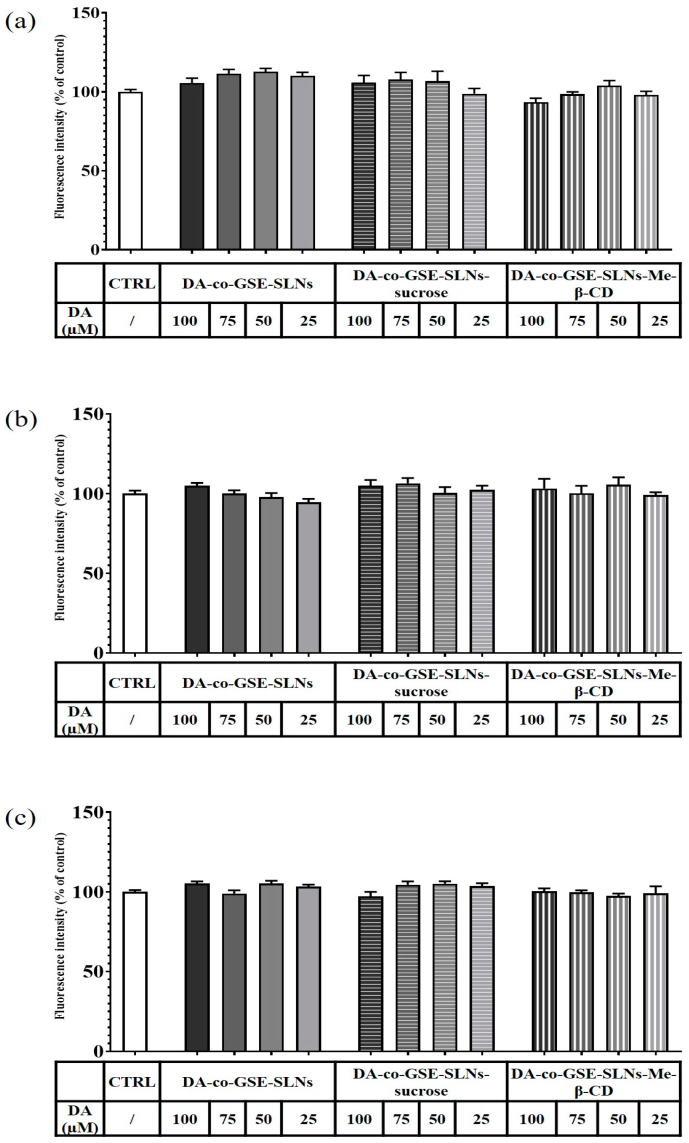
Resazurin viability assay on RPMI 2650 cells after 6 h (**a**), 12 h (**b**) and 24 h (**c**) treatment with DA-co-GSE-SLNs, DA-co-GSE-SLNs-sucrose and DA-co-GSE-SLNs-Me-β-CD. DA concentration was in the range of 25–100 µM. In all the experimental conditions, cell viability did not significantly change compared to the control (untreated cells). *p* > 0.05 not significant.

**Figure 10 molecules-28-07706-f010:**
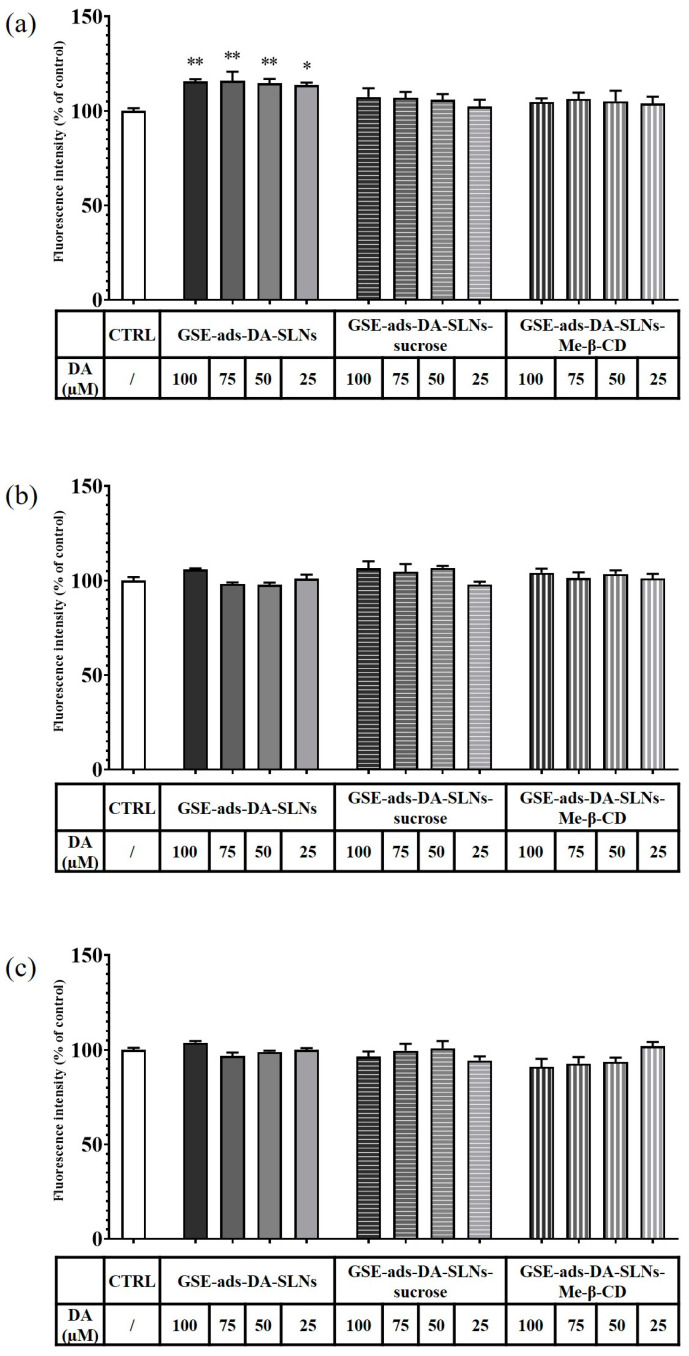
Resazurin viability assay on RPMI 2650 cells after 6 h (**a**), 12 h (**b**) and 24 h (**c**) treatment with GSE-ads-DA-SLNs, GSE-ads-DA-SLNs-sucrose and GSE-ads-DA-SLNs- Me-β-CD. DA concentration was in the range of 25–100 µM. Statistical analysis using ANOVA and Dunnett post hoc test: 6 < n < 9 for all the treatments; * *p* < 0.05; ** *p* < 0.01. The absence of asterisks indicates no significant data (*p* > 0.05).

**Table 1 molecules-28-07706-t001:** Physicochemical properties of SLNs with or without cryoprotectants. Mean ± standard deviation of at least eight replicates is reported. For size and zeta potential statistical evaluation, each formulation before freeze-drying was taken as control.

Formulation	Size (nm)	PDI ^a^	Zeta Potential (mV)	E.E. DA (%)	E.E. GSE (%)	D.L. DA (%)	D.L. GSE (%)
DA-co-GSE-SLNs ^b^	187 ± 4	0.49 ± 0.04	−4.1 ± 0.1	62 ± 4	10 ± 0	0.10 ± 0.006	0.01 ± 0
DA-co-GSE-SLNs b. f. d ^c^	456 ± 142	0.61 ± 0.10	−16.5 ± 1.1	N.D.	N.D.	N.D.	N.D.
DA-co-GSE-SLNs a. f. d.^d^	583 ± 180	0.44 ± 0.20	−21.4 ± 2.2	62 ± 4	10 ± 0	0.10 ± 0	0.01 ± 0
DA-co-GSE-SLNs-Me-β-CD b. f. d.	317 ± 61	0.58 ± 0.10	−18.7 ± 2.4				
DA-co-GSE-SLNs-Me-β-CD a. f. d.	614 ± 56 *	0.67 ± 0.62	−6.9 ± 0.5 **	50 ± 2	9 ± 1	0.08 ± 0.003	0.009 ± 0.001
DA-co-GSE-SLNs-sucrose b. f. d.	469 ± 4	0.70 ± 0.05	−36.0 ± 0.8				
DA-co-GSE-SLNs-sucrose a. f. d.	424 ± 54	0.84 ± 0.24	−29.4 ± 3.9 **	46 ± 7	8 ± 0.4	0.07 ± 0.012	0.008 ± 0.0004
GSE-ads-DA-SLNs ^b^	287 ± 15	0.53 ± 0.01	7.8 ± 0.4	65 ± 6	57 ± 8	0.22 ± 0.022	0.0095 ± 0.001
GSE-ads-DA-SLNs b. f. d. ^c^	685 ± 38	0.59 ± 0.14	−20.9 ± 1.6	N.D.	N.D.	N.D.	N.D.
GSE-ads-DA-SLNs a. f. d.	764 ± 95	0.81 ± 0.27	−18.9 ± 1.1	65 ± 6	57 ± 8	0.22 ± 0.022	0.0095 ± 0.001
GSE-ads-DA-SLNs-Me-β-CD b. f. d	206 ± 94	0.43 ± 0.13	−13.6 ± 1.3				
GSE-ads-DA-SLNs-Me-β-CD a. f. d.	342 ± 74	0.70 ± 0.19	−10.3 ± 0.6	45 ± 3	40 ± 4	0.15 ± 0.016	0.006 ± 0.0006
GSE-ads-DA-SLNs-sucrose b. f. d.	431 ± 99	0.34 ± 0.0	−26.2 ± 3.9				
GSE-ads-DA-SLNs-sucrose a. f. d.	≥1000 **	0.67 ± 0.13	−20.6 ± 5.9	69 ± 6	52 ± 6	0.23 ± 0.024	0.008 ± 0.0006

^a^ PDI: polydispersity index. ^b^ Particle size of freshly prepared nanosuspensions without any cryoprotectant and freeze-drying cycle (from Ref. [12]).^c^ Before freeze-drying (b. f. d.) without protectant and after two weeks of storage at 4 °C in the refrigerator. ^d^ After freeze-drying (a. f. d.). * *p* ≤ 0.05; ** *p* ≤ 0.01.

**Table 2 molecules-28-07706-t002:** Raman intensity ratios related to C-C stretching vibrational bands and to C-H stretching vibrational bands of the investigated SLNs after freeze-drying.

Sample	I_2890_/I_2850_	I_1115_/I_1050_
Gelucire^®^ 50/13	1.44	1.06
Plain SLNs	1.41	1.02
DA-co-GSE-SLNs	1.13	-
GSE-ads-DA-SLNs	1.12	0.99
DA-co-GSE-SLNs-sucrose	-	1.12
DA-co-GSE-SLNs-Me-β-CD	1.57	1.28
GSE-ads-DA-SLNs-sucrose	-	1.12
GSE-ads-DA-SLNs-Me-β-CD	1.55	1.19

**Table 3 molecules-28-07706-t003:** Surface atomic composition of the freeze-dried formulations examined by XPS.

Formulation	C1s	O1s	N1s	Si2p	Cl2p
DA-co-GSE-SLNs-Me-β-CD	66.0	34.0	--	--	--
DA-co-GSE-SLNs-sucrose	79.9	17.5	0.4	2.3	--
GSE-ads-DA-SLNs-Me-β-CD	76.0	22.1	0.2	1.6	--
GSE-ads-DA-SLNs-sucrose	63.7	32.4	--	3.9	--
DA-co-GSE-SLNs	84.3	14.4	0.7	--	0.6
GSE-ads-DA-SLNs	85.4	13.6	0.5	--	0.4

## Data Availability

The datasets generated during the current study are available from the corresponding author upon reasonable request.

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
