# Peer review of "Solid Lipid Nanoparticles Containing Dopamine and Grape Seed Extract: Freeze-Drying with Cryoprotection as a Formulation Strategy to Achieve Nasal Powders"

_molecules, 2023, doi:10.3390/molecules28237706_

Round 1
Reviewer 1 Report
Comments and Suggestions for Authors
The manuscript by Trapani et al. describes the effect of two cryoprotectants on the morphology and structure of two type of solid lipid nanoparticles (SLNs) that are ultimately intended for the intranasal delivery of dopamine (DA) and grape seed extract (GSE) to the central nervous system as a treatment for Parkinson’s disease.
The works has some merits but the overall impression is that some information is missing and that the conclusions are not supported by the experimental results. The authors’ conclusion on the effect of freeze-drying on the size of the SLNs in not clear. The major impact seems to be that of the protectants themselves, rather than freeze-drying…
General concerns:
Then the authors claim that the higher drug release from DA-co-GSE-SLNs-Me-β-CD SLNs is explained by the presence of pores, which is only supported by AFM (only shown in sup info) that basically only shows that the surface of these SLNs is rough. This not enough to prove the presence of pores. Specific experiments (BET?) should be undertaken to confirm the porous nature of these SLNs.
The “folded sheets” shown in SEM analysis are of very little interest. The scale of this analysis seems irrelevant.
What the authors conclude from their Raman spectroscopy study is unclear. It seems that just conclude that simulated nasal fluid has an impact on the spectra and… that’s it. This should be better discussed in order to highlight what this study provided in terms of understanding the interaction between SLNs and cryoprotectants (which is the topic of the article).
XPS analyses are only moderately informative and leads to the logical conclusion that only outer components can be probed by the technique, without much insight on the problem at hand.
Even though the authors claim otherwise (“DA-co-GSE-SLNs and GSE-ads-DA-SLNs reduced cell viability in comparison to the absence of Me-β-CD at 12and24h of treatment (Figures 9b and 9c; Figures 10b and 10c)”), there seems to be no impact of the presence of either cryoprotectants on cell survival, based on Figures 9 and 10. This should be clarified.
Specific concerns:
Table 1: How can the authors explain the size difference between lines 1&2 (or 8&9)? Line 1 is the freshly prepared particles and line 2 is the same particles before freeze-drying without protectant. So they are basically the same. However, there is a huge size difference. This is unclear and needs to be commented.
In Figure 1, why is there no PCS profile for SLNs in the presence of either cryoprotectant but before freeze-drying? These should be shown, described and commented.
The authors seem to be confused as to whether they should use “drug loading” or “E.E.” (encapsulation efficacy). I do not understand how they can calculate an EE for particles before and after freeze-drying since the encapsulation is done prior to f-d anyways. E.E. is determined after the encapsulation step, period. If the drug content decreased after freeze-drying, it has nothing to do with EE, it’s simply leaking/degradation. It would be more adequate to calculate drug loadings instead (expressed in wt%).
There are no error bars in Figure 2 a) and b). These should be added in order to better illustrate the significance of the differences of release profiles.
How many particles were measured for AFM size analysis?
The discussion section is very difficult to follow. It is long but does not provide much definitive insight on the results. The authors should rethink the whole section and present a more concise discussion that focusses on relevant and significant findings.
In the discussion, the authors say that there is a difference in cell viability when CD is used as protectant as opposed to sucrose or without protectant. However, I fail to see any significant difference in Figures 9 or 10 regarding the use of CD.
Figures 9 and 10 are too small to be readable. There is no statistics on panels b and c. Does that mean that no significant difference is observed. It should be noted (ns) on the figures.
As it stands, I cannot recommend publication of this article in Molecules. The authors should address the above-mentioned concerns and resubmit their manuscript.
Comments on the Quality of English LanguageThere are numerous typos/errors (repeated words, grammar errors, comments between authors that were not removed before submission, etc.) throughout the manuscript. Many sentences sound like Italian translated into English word for word. This manuscript must be thoroughly edited by a native English speaker.
Author Response
Reply to Editor and Reviewer comments:
Editor
Please check that all references are relevant to the contents of the manuscript.
We checked that all references are relevant to the content of the manuscript
We propose that you use one of the editing services listed at https://www.mdpi.com/authors/english or have your manuscript checked by a colleague fluent in English writing.
According to the Editor suggestion, an in-depth check of the English language has been performed by a colleague fluent in English writing.
Reviewer 1
The works has some merits but the overall impression is that some information is missing and that the conclusions are not supported by the experimental results. The authors’ conclusion on the effect of freeze-drying on the size of the SLNs in not clear. The major impact seems to be that of the protectants themselves, rather than freeze-drying…
We thank the Reviewer for his/her opinion about our manuscript and for the comments aimed at its improvement. In the revised version, we have clearly stated in Conclusion section that the whole freeze-drying cycle using sucrose as cryoprotectant is advantageous in the improvement of SLN stability. In addition, also the title of the manuscript has been slightly changed inserting: ”… freeze-drying with cryoprotection as formulation strategy to achieve nasal powders”. We believe that, in such a way, the problem pointed out by the Reviewer (i.e., that “The major impact seems to be that of the protectants themselves, rather than freeze-drying) should be overcome.
General concerns:
- Then the authors claim that the higher drug release from DA-co-GSE-SLNs-Me-β-CD SLNs is explained by the presence of pores, which is only supported by AFM (only shown in sup info) that basically only shows that the surface of these SLNs is rough. This not enough to prove the presence of pores. Specific experiments (BET?) should be undertaken to confirm the porous nature of these SLNs.
- We acknowledge the Reviewer for the interesting and challenging question. We have given much thought to the problem of characterizing particle porosity, but we haven't found a solution yet. The samples must be thoroughly dried in all conventional methods for porosity analysis (e.g., BET or Mercury porosimetry). For this purpose, only classic techniques such as lyophilization (described in this publication) are feasible for lipid nanoparticles. However, these drying processes already influence the porosity. So, how do you determine the "initial value"? In addition, the geometry of the lipid particles is affected by the methods (e.g., BET), too. Lipid particles have a high compressibility and are very soft. Thus, they do not tolerate methods such as BET, Mercury porosimetry, or "Beckmann".
- The “folded sheets” shown in SEM analysis are of very little interest. The scale of this analysis seems irrelevant.
- In the revised version, morphological observations concerning SLNs carried out by SEM and AFM microscopies were shown in the same section renumbered as 2.3.3. In Discussion section, we have indicated that both techniques are complementary each other in the interpretation of SLN surface features, by adding a new sentence “Indeed, the SEM shows the surface morphology of the composite in as-prepared conditions, whereas the AFM investigated that of the individual SLNs, after isolating them through the ad-hoc procedure devised and described in our paper.”
- What the authors conclude from their Raman spectroscopy study is unclear. It seems that just conclude that simulated nasal fluid has an impact on the spectra and… that’s it. This should be better discussed in order to highlight what this study provided in terms of understanding the interaction between SLNs and cryoprotectants (which is the topic of the article).
- In the revised section 2.3.2, we have listed vibrations of covalent bonds in the sucrose molecule due to the reciprocal interactions between SLNs and cryoprotectant. From the analysis of Raman spectra, it’s possible to see peak shift and intensity change of the SLNs with cryoprotectants which is related to the interactions existing between SLNs and cryoprotectant. Indeed, in the Raman spectra, shifting of peaks towards lower or higher wavenumber is related to chemical bond length of molecules. The shortening of bond length causes shift to higher wavenumber. If the chemical bond length of a molecule changes due to any internal or external effects (like the SNF exposure to the SNF solvent), then a wavenumber shift is recorded.
- XPS analyses are only moderately informative and leads to the logical conclusion that only outer components can be probed by the technique, without much insight on the problem at hand.
- Concerning the Reviewer's issue, XPS is a surface technique that allows examining a maximum thickness of 10 nm of a solid sample. In the literature, this technique is often used to examine the surface composition of SLNs, consisting of several components, to highlight the surface presence of one rather than another (https://doi.org/10.1016/j.msec.2013.01.037,https://doi.org/10.1016/j.biomaterials.2011.01.048, https://doi.org/10.1016/j.ijpharm.2014.06.022, https://doi.org/10.1016/j.ijpharm.2017.06.045, https://doi.org/10.1016/j.jddst.2018.08.013 etc.). Therefore, in our opinion, XPS may provide useful information on the surface composition of all examined SLNs as reported in the manuscript. In particular, this analysis highlighted in some systems the superficial presence of the neurotransmitter and/or the cryoprotectant which, in our case, is of significant importance.
- Even though the authors claim otherwise (“DA-co-GSE-SLNs and GSE-ads-DA-SLNs reduced cell viability in comparison to the absence of Me-β-CD at 12 and 24 h of treatment (Figures 9b and 9c; Figures 10b and 10c)”), there seems to be no impact of the presence of either cryoprotectants on cell survival, based on Figures 9 and 10. This should be clarified.
- We acknowledge the Reviewer for the issue that in Figure 9 none of the results shown are significant. We have consequently corrected the text in Section 2.4 and we have also edited accordingly Figure 9 caption.
Specific concerns:
- Table 1: How can the authors explain the size difference between lines 1&2 (or 8&9)? Line 1 is the freshly prepared particles and line 2 is the same particles before freeze-drying without protectant. So they are basically the same. However, there is a huge size difference. This is unclear and needs to be commented.
- In the revised form, in Table 1 we make it clear that lines 2 and 9 refer to freshly prepared SLNs which were freeze dried without protectant after two weeks of storage at 4 °C in the refrigerator. Hence, the huge size difference pointed out by the Reviewer may be due to physical instability of the corresponding nanoparticles during the two weeks of storage at 4°C in the refrigerator without protectant. In Discussion Section we have inserted a sentence accounting for the difference evidenced by the Reviewer. We thank the Reviewer for this comment since it allowed us to clarify this inconsistency.
- In Figure 1, why is there no PCS profile for SLNs in the presence of either cryoprotectant but before freeze-drying? These should be shown, described and commented.
- Unfortunately, we did not record the PCS profiles of SLNs in the presence of either cryoprotectant but before freeze drying and, therefore, we cannot satisfy the Reviewer’s request. Sorry for this.
- The authors seem to be confused as to whether they should use “drug loading” or “E.E.” (encapsulation efficacy). I do not understand how they can calculate an EE for particles before and after freeze-drying since the encapsulation is done prior to f-d anyways. E.E. is determined after the encapsulation step, period. If the drug content decreased after freeze-drying, it has nothing to do with EE, it’s simply leaking/degradation.
- R. According to the Reviewer’s request, in the revised version in section 4.3 we have inserted drug loading equation and, hence, Table 1 was also edited accordingly. Overall, we can rule out any neurotransmitter degradation process since we did not observe any HPLC DA signal modification after freeze-drying with cryoprotection.
- There are no error bars in Figure 2 a) and b). These should be added in order to better illustrate the significance of the differences of release profiles.
- Standard deviations measured for the data points in Figure 2a,b were very small and, therefore this is why no error bars appear in the mentioned Figures. This information has been added in footnote of Figure 2.
- How many particles were measured for AFM size analysis?
- In section 4.9 of the revised manuscript, we have clarified that we have evaluated 75 particles each to determine the particle size via AFM.
- The discussion section is very difficult to follow. It is long but does not provide much definitive insight on the results. The authors should rethink the whole section and present a more concise discussion that focusses on relevant and significant findings.
- In the resubmitted manuscript, we have shortened the Discussion section according to the Reviewer’s request.
- In the discussion, the authors say that there is a difference in cell viability when CD is used as protectant as opposed to sucrose or without protectant. However, I fail to see any significant difference in Figures 9 or 10 regarding the use of CD.
- The Reviewer’s observation led us to edit the Discussion Section, introducing the sentences: “Similarly, the addition of Me-β-CD as a cryoprotectant to both DA-co-GSE-SLNs and GSE-ads-DA-SLNs determined only a very slight, not significant reduction in cell viability in comparison to the absence of Me-β-CD, irrespectively of the time of treatment (Figures 9b and 9c; Figures 10b and 10c). To account for this very low tendency reduction in cell viability observed when Me-β-CD was used as cryoprotectant, the mentioned biological membrane non-permeable feature of this methylated CD could be invoked”.
- Figures 9 and 10 are too small to be readable. There is no statistics on panels b and c. Does that mean that no significant difference is observed. It should be noted (ns) on the figures.
- We thank the Reviewer for this comment. We have enlarged both Figures 9 and 10 and we think they are now much more readable and clear. Regarding statistics, we have edited both captions to Figure 9 and 10, reporting that data are not significant for p values>0.05. Given that in Figures 9 and 10 many values are not significant, in order tp prevent that figures look too busy, we would like to avoid to cite“ns” on their bargraphs.
- Comments on the Quality of English Language. There are numerous typos/errors (repeated words, grammar errors, comments between authors that were not removed before submission, etc.) throughout the manuscript. Many sentences sound like Italian translated into English word for word. This manuscript must be thoroughly edited by a native English speaker.
- We apologize for the mistakes related to the English language occurred. In the resubmitted manuscript, an in-depth check of the English language has been performed throughout the text by a colleague fluent in English writing.
Reviewer 2 Report
Comments and Suggestions for Authors
In the manuscript entitled “Solid Lipid Nanoparticles containing Dopamine and Grape Seed Extract: cryoprotection as formulation strategy to achieve nasal powders” authors developed solid lipid nanoparticles loaded with dopamine and grape seed extract for a possible treatment of Parkinson’s disease and tested two different cryoprotectant agents.
The work presented by De Giglio and collaborators is interesting and well fits with the selected special issue. However, minor revisions should be made to be published in Molecules journal.
- The influence of grape seed extract in the treatment of PD should be better explained. Thus, authors cite its antioxidant activity, but a deeper explanation could be useful to explain its selection (for example, see Tsarouchi, M.; Fanarioti, E.; Karathanos, V.T.; Dermon, C.R. Protective Effects of Currants (Vitis vinifera) on Corticolimbic Serotoninergic Alterations and Anxiety-like Comorbidity in a Rat Model of Parkinson’s Disease. Int. J. Mol. Sci. 2023, 24, 462. https://doi.org/10.3390/ijms24010462)
- The exploited GSE is described as a mixture of more proanthocyanidins. However, the described HPLC chromatogram reports only one peak for GSE. Please provide a representative chromatogram (in the main text or in the supplementary) and provide explanation about this point.
- Have you considered the stability of proanthocyanidins in acidic pH?
Comments on the Quality of English LanguageEnglish editing is required. Particularly, lines 142, 273,379-380, 331-332 present typo or incorrect sentences.
Author Response
Reviewer 2
Comments and Suggestions for Authors
In the manuscript entitled “Solid Lipid Nanoparticles containing Dopamine and Grape Seed Extract: cryoprotection as formulation strategy to achieve nasal powders” authors developed solid lipid nanoparticles loaded with dopamine and grape seed extract for a possible treatment of Parkinson’s disease and tested two different cryoprotectant agents.
The work presented by De Giglio and collaborators is interesting and well fits with the selected special issue. However, minor revisions should be made to be published in Molecules journal.
We thank the Reviewer for his/her overall positive opinion about our work
- The influence of grape seed extract in the treatment of PD should be better explained. Thus, authors cite its antioxidant activity, but a deeper explanation could be useful to explain its selection (for example, see Tsarouchi, M.; Fanarioti, E.; Karathanos, V.T.; Dermon, C.R. Protective Effects of Currants (Vitis vinifera) on Corticolimbic Serotoninergic Alterations and Anxiety-like Comorbidity in a Rat Model of Parkinson’s Disease. Int. J. Mol. Sci. 2023, 24, 462. https://doi.org/10.3390/ijms24010462).
- We thank the Reviewer for the suggestion to better point out the effect of grape seed extract in the treatment of PD. For this purpose, in the Introduction section of the revised version, we inserted a sentence in this regard.
- The exploited GSE is described as a mixture of more proanthocyanidins. However, the described HPLC chromatogram reports only one peak for GSE. Please provide a representative chromatogram (in the main text or in the supplementary) and provide explanation about this point.
- In the Supplementary Material we show the HPLC chromatogram corresponding to GSE used in the present work. As can be seen it results, an intense peak due to a mixture of polyphenols (i.e., proanthocyanidins) of comparable lipophilicity together with a little peak probably due to further substances of non-proanthocyanidin nature. Based on this HPLC profile, we can state that the content in proanthocyanidins of Grape Seed Extract used by us was ≥ 95.0% as reported in section 4.1 of the text.
- Have you considered the stability of proanthocyanidins in acidic pH?
- It should be taken into consideration that the neurotransmitter Dopamine (DA) is a very sensitive substance which undergo a prompt autoxidation reaction particularly in alkaline medium (Umek et al Dopamine Autoxidation Is Controlled by Acidic pH, Front. Mol. Neurosci (2018) 11:467. doi: 10.3389/fnmol.2018.00467) leading to a nanosuspension grayish colored due to polymeric compound formation. Hence, we are obliged to use DA in acidic conditions (pH = 6) to control the autoxidation reaction. Under such conditions, we did not observe in our hands formation of autoxidation products nor changes in color of nanosuspensions as well as it is unchanged the peak corresponding to DA in HPLC. Similarly, the peak corresponding to GSE in HPLC did not show any change in the acidic conditions used nor there were additional peaks attributable to degradation of GSE. Therefore, we conclude that, under the conditions used, GSE is a stable substance.
Round 2
Reviewer 1 Report
Comments and Suggestions for Authors
The authors have done a decent job of responding to most of my admittedly numerous comments. The writing isn't polished enough, and there are still grammatical and typographical errors, even in some headings (see section 4.5, for example). I still find the article a little lacking, but I'll leave the final decision to the editor.
Comments on the Quality of English Languagetoo many errors remain